# Off-Diagonal Magnetoimpedance in Annealed Amorphous Microwires with Positive Magnetostriction: Effect of External Stresses

Nikita A. Buznikov 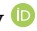

Scientific and Research Institute of Natural Gases and Gas Technologies–Gazprom VNIIGAZ, 142717 Razvilka, Moscow Region, Russia; n_buznikov@mail.ru

**Abstract:** It was observed recently that the giant magnetoimpedance (GMI) effect in Fe-rich glass-coated amorphous microwires with positive magnetostriction can be improved significantly by means of post-annealing. The increase in the GMI is attributed to the induced helical magnetic anisotropy in the surface layer of the microwire, which appears after the annealing. The application of external stresses to the microwire may result in changes in its magnetic structure and affect the GMI response. In this work, we study theoretically the influence of the tensile and torsional stresses on the off-diagonal magnetoimpedance in annealed amorphous microwires with positive magnetostriction. The static magnetization distribution is analyzed in terms of the core–shell magnetic structure. The surface impedance tensor is obtained taking into account the magnetoelastic anisotropy induced by the external stresses. It is shown that the off-diagonal magnetoimpedance response exhibits strong sensitivity to the magnitude of the applied stress. The obtained results may be useful for sensor applications of amorphous microwires.

**Keywords:** magnetoimpedance; amorphous microwires; annealing; induced anisotropy; magnetostriction; magnetoelastic anisotropy





## 1. Introduction

The giant magnetoimpedance (GMI) effect consists of a significant change in the impedance of a soft magnetic conductor upon the application of an external magnetic field. The GMI can be explained by the skin effect, and the field dependence of the impedance is attributed to changes in the permeability with the external magnetic field. The GMI was intensively studied in various soft magnetic amorphous and nanocrystalline materials (see, for example, [1–3] and references therein). The high field sensitivity of the GMI attracts considerable attention due to its possible use in different technological applications, in particular, for the development of biosensors, position sensors, magnetometers, systems for non-destructive testing, etc.

Glass-coated amorphous microwires are one of the most promising materials for applications of GMI. These microwires are produced by Taylor–Ulitovsky method and represent composite materials consisting of a metallic nucleus with a diameter of 1–50 μm covered by glass with a thickness of 2–20 μm [4–8]. Strong GMI effects are observed in Co-rich amorphous microwires with nearly-zero magnetostriction [7,8]. The specific distribution of the residual stresses emerging during fabrication leads to the appearance of circular or helical magnetic anisotropy in the microwires [5,9]. As a result, Co-rich glass-coated amorphous microwires exhibit high values of transverse permeability resulting in the strong GMI effect [1,2].

Another type of microwire is Fe-rich glass-coated amorphous microwire with positive magnetostriction. The residual stresses result in a fundamentally different distribution of the magnetic anisotropy in Fe-rich microwires in comparison with Co-rich microwires.

Fe-rich amorphous microwires have predominately axial anisotropy, low transverse permeability, and, consequently, possess a weak GMI effect. However, recently it has been demonstrated that post-annealing of Fe-rich glass-coated amorphous microwires may enhance significantly the GMI effect. In particular, the magnetic softness and GMI can be improved by means of stress-annealing [10–14]. It was observed that the GMI ratio increases by an order of magnitude after stress-annealing [11,12]. It was found also that the GMI field dependence changes drastically with an increase in frequency, transforming from single-peak behavior to two-peak dependence. A similar increase in the GMI effect has been observed recently in Fe-rich glass-coated amorphous microwires after Joule heating [15–17].

A theoretical description of the changes in magnetic properties of stress-annealed glass-coated amorphous microwires with positive magnetostriction has been proposed [18]. It was shown that the heating of the microwire in the presence of the tensile stress changes the distribution of the residual stresses in the metallic amorphous nucleus. After the stress-annealing, the tangential residual stresses in the surface region of the amorphous nucleus become maximum, which results in the appearance of circular anisotropy in this region [18].

To describe the GMI effect in stress-annealed Fe-rich amorphous microwires an electrodynamic model has been developed [19]. In the framework of the model, the static magnetization distribution within the metallic part of the microwire was described by means of the core–shell magnetic structure, assuming the existence of the inner core with an axial magnetic anisotropy and external shell having a helical anisotropy. The model allows one to describe qualitatively the experimental GMI field and frequency dependences in field-annealed glass-coated amorphous microwires with positive magnetostriction [10–14].

From the point of view of sensor applications, much attention is paid to off-diagonal magnetoimpedance (ODMI). The ODMI consists of the appearance of the field-dependent voltage in the pick-up coil wound around a conductor and is attributed to the cross-magnetization process [20,21]. The studies of the ODMI in Co-rich glass-coated amorphous microwires show that the pick-up coil voltage signal is very small and irregular [22,23]. This fact is attributed to the so-called bamboo domain structure in the surface region of the microwire. To eliminate the effect of the domain structure, the direct bias current is used, which leads to the asymmetric field dependence of the ODMI. The asymmetric ODMI response is promising for applications due to its linear field dependence and enhanced sensitivity. It was found also that a magnetometer based on the asymmetric ODMI effect exhibits better noise performance in comparison with the GMI [24,25].

The ODMI was investigated also in Fe-rich amorphous microwires [11–13]. In as-prepared Fe-rich amorphous microwires, the observed ODMI response was very small. After stress-annealing, the ODMI increases, and it can be enhanced further by using the bias current [11–13].

Since the domain structure and GMI response in an amorphous sample strongly depend on the residual stress distribution, the application of external stresses may change significantly the conductor impedance. The phenomenon is often referred to as the stress impedance effect [26–28]. The stress impedance could be promising for the development of stress and strain sensors. Furthermore, the application of stresses to an amorphous conductor leads to changes in the domain structure and, hence, to variations in the GMI. The influence of the tensile and torsional stresses on the GMI effect in Co-rich glass-coated amorphous microwires have been intensively studied (see, for example, [29–33]). At the same time, the effect of external stresses on the GMI in annealed Fe-rich amorphous microwires remains completely unexplored.

In this work, we present a theoretical study of the influence of the tensile and torsional stresses on the ODMI in annealed glass-coated amorphous microwires with positive magnetostriction. The static magnetization distribution is described based on the core–shell magnetic structure taking into account the magnetoelastic anisotropy induced by the external stresses. The expressions for the microwire impedance tensor are found through the

values of the transverse permeability in the core and shell. The ODMI effect is analyzed as a function of the external field, frequency, and magnitude of the applied stress.

## 2. Model

### 2.1. Static Magnetization Distribution

Following the approach developed previously in [19], let us assume that a glass-coated amorphous microwire with the diameter of metallic nucleus 2*R* consists of two regions with different types of magnetic anisotropy. In the inner region (core) with the diameter 2*r*, the anisotropy is axial, and the outer region (shell) has a helical anisotropy induced after stress-annealing. The microwire is subjected to the alternating current $I(t) = I_0 \exp(-i\omega t)$ (here *t* is the time, $\omega$ is the angular frequency and *i* is the imaginary unit) and to the direct bias current $I_b$, and the external magnetic field $H_e$ is parallel to the microwire axis. The tensile stress $\sigma_t$ and uniform torsional stress with the angular displacement per unit length $\tau$ are applied to the microwire. The microwire cross-section and a sketch of the coordinate system used for analysis are shown in Figure 1. Since all fields lie within the $\varphi$–*z* plane, the magnetization vector is limited to the same plane.

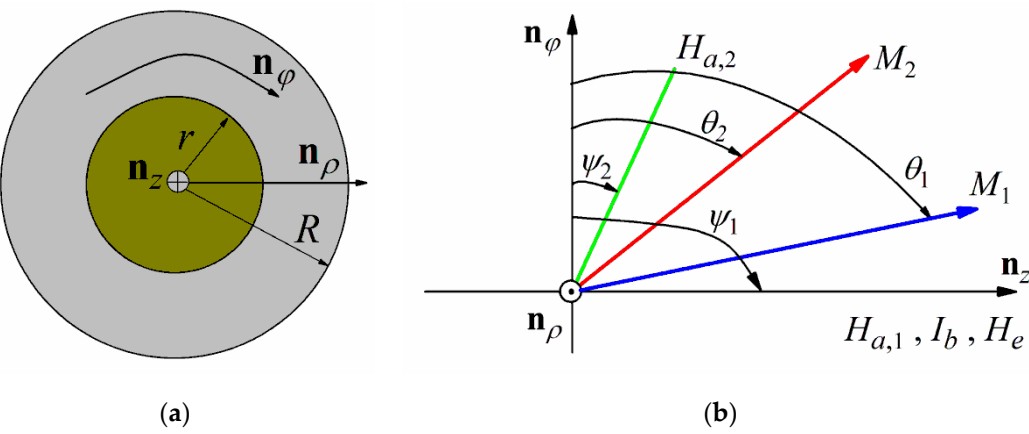

(**a**) (**b**)

**Figure 1.** (**a**) The microwire cross-section and unit vectors $\mathbf{n}_\rho$, $\mathbf{n}_\varphi$ and $\mathbf{n}_z$ of the coordinate system used for analysis. (**b**) A sketch of the angles in the model lying within the $\mathbf{n}_\varphi$–$\mathbf{n}_z$ plane.

Further, we assume that the magnetization is uniform within two regions of the microwire, and the exchange and magnetostatic coupling between the regions can be neglected [19]. Note that it was demonstrated that account of the exchange coupling between the core and shell regions can improve an agreement between theory and experiment for the GMI response in amorphous wires [34]. As a matter of principle, the coupling can be taken into account by introducing the effective interaction field between the core and shell [32,34]. However, in this case, the model becomes more complicated. In this regard, to simplify the model we neglect the coupling between the core and shell regions of the microwire.

The static equilibrium magnetization distribution within the core and shell can be obtained by the minimization of the free energy. The free energy density *U* can be presented as a sum of the intrinsic magnetic anisotropy term, Zeeman energy, and magnetoelastic anisotropy energy induced by the external stresses:

$$U = (MH_{a,j}/2)\sin^2(\theta_j - \psi_j) - MH_e \sin\theta_j + U_b + U_{\text{ten}} + U_{\text{tor}}. \tag{1}$$

Here and hereafter the subscript *j* = 1 and *j* = 2 corresponds to the core and shell region, respectively; *M* is the saturation magnetization; $H_{a,j}$ is the intrinsic magnetic anisotropy fields in the core and shell; $\theta_j$ and $\psi_j$ are the equilibrium magnetization angles and the anisotropy axis angles with respect to the azimuthal direction ($\psi_1 = \pi/2$ for the core region); $U_b$ is the Zeeman energy of the field of the bias current; $U_{\text{ten}}$ and $U_{\text{tor}}$ are the magnetoelastic anisotropy energies induced by the tensile and torsional stresses.

The term $U_b$ can be written as follows

$$U_b = -2MI_b\rho \cos\theta_j / cR^2,$$ (2)

where $\rho$ is the radial coordinate and $c$ is the speed of light in a vacuum.

The magnetoelastic anisotropy energy $U_{\text{ten}}$ induced by the tensile stress has the following form [1]:

$$U_{\text{ten}} = (3/2)\lambda_s\sigma_t \cos^2\theta_j,$$ (3)

where $\lambda_s > 0$ is the magnetostriction coefficient.

The torsional stress results in the appearance of the helical magnetoelastic anisotropy in the microwire, with the easy magnetization axis being the angle of $\pi/4$ with azimuthal direction. The magnetoelastic anisotropy energy $U_{\text{tor}}$ can be expressed as [1]

$$U_{\text{tor}} = (3/2)\lambda_s G\tau\rho \sin^2(\theta_j - \pi/4),$$ (4)

where $G$ is the shear modulus.

It follows from Equations (2) and (4) that the Zeeman energy $U_b$ of the bias field and the magnetoelastic anisotropy $U_{\text{tor}}$ due to the torsional stress varies over the microwire cross-section. To simplify the model, we assume further that the contributions of these terms to the free energy density corresponding to the maximal values for the core (at $\rho = r$) and shell (at $\rho = R$) regions.

Then, the minimization of the free energy density results in the following equations for the equilibrium magnetization angles $\theta_j$ in the core and shell:

$$
\begin{aligned}
&H_{a,j}\sin(\theta_j - \psi_j)\cos(\theta_j - \psi_j) - H_e\cos\theta_j + H_{b,j}\sin\theta_j \\
&- (H_{\text{ten}}/2)\sin 2\theta_j - (H_{\text{tor},j}/2)\cos 2\theta_j = 0 \,.
\end{aligned}
$$ (5)

Here we introduce the fields

$$H_{b,1} = 2I_b r / cR^2,$$ (6)

$$H_{b,2} = 2I_b / cR,$$ (7)

$$H_{\text{ten}} = 3\lambda_s\sigma_t / M,$$ (8)

$$H_{\text{tor},1} = 3\lambda_s G\tau r / M,$$ (9)

$$H_{\text{tor},2} = 3\lambda_s G\tau R / M.$$ (10)

Note that the general approach to describe the $U_b$ and $U_{\text{tor}}$ energy terms consists of the integration of the free energy density over the radial coordinate and the following minimization of the resulting energy. Since both the $U_b$ and $U_{\text{tor}}$ terms depend linearly on the radial coordinate, the minimization procedure leads to Equation (2) for the equilibrium magnetization angles, and the only difference is in the numerical coefficients in Equations (6), (7), (9) and (10). Thus, the simplified model described above gives the same basic predictions as compared to the general approach. In addition, the assumption that the contributions of the $U_b$ and $U_{\text{tor}}$ terms correspond to their maximal values in both regions allows one to describe the conditions for the fields at the surface of the metallic nucleus of the microwire.

It follows from Equation (5) that the magnetoelastic anisotropy interacts with the intrinsic magnetic anisotropy and changes the equilibrium magnetization angles $\theta_j$. Equation (5) can be rewritten in the following form with effective uniaxial anisotropy:

$$H_{\text{eff},j}\sin(\theta_j - \alpha_j)\cos(\theta_j - \alpha_j) - H_e\cos\theta_j + H_{b,j}\sin\theta_j = 0.$$ (11)

Here $H_{\text{eff},j}$ and $\alpha_j$ are the effective anisotropy fields and anisotropy angles, correspondingly. After simple mathematical transformations, for $\alpha_j$ and $H_{\text{eff},j}$ we obtain [31]:

$$\tan 2\alpha_j = \frac{H_{a,j}\sin 2\psi_j + H_{\text{tor},j}}{H_{a,j}\cos 2\psi_j - H_{\text{ten}}},\tag{12}$$

$$H_{\text{eff},j} = \frac{H_{a,j}\cos 2\psi_j - H_{\text{ten}}}{\cos 2\alpha_j}.\tag{13}$$

Let us discuss the changes in the effective anisotropy fields and angles in the presence of external stresses. The application of the tensile stress $\sigma_t$ (at $\tau = 0$) increases the axial effective anisotropy field $H_{\text{eff},1}$ in the core region. It follows from Equations (12) and (13) that for the shell region, the growth of $\sigma_t$ leads to a deviation of the effective anisotropy angle $\alpha_2$ from the azimuthal direction and a decrease of the effective anisotropy field $H_{\text{eff},2}$. Note that these dependencies are opposite to the behavior of the effective anisotropy in Co-rich amorphous microwires with slightly negative magnetostriction. It is well-known that in Co-rich microwires, the effective anisotropy axis rotates toward the azimuthal direction with a growth of the tensile stress, and the effective anisotropy field increases [1,30–32].

When the torsional stress is applied to the microwire (at $\sigma_t = 0$), the angle $\alpha_1$ in the core region changes within the range from $3\pi/4$ at the high negative direction of the stress (counter-clockwise rotation) to $\pi/4$ at high positive stress (clockwise rotation). The effective anisotropy field $H_{\text{eff},1}$ increases monotonically with the stress. Figure 2 illustrates the influence of the torsional stress on the effective anisotropy angle $\alpha_2$ and the effective anisotropy field $H_{\text{eff},2}$ in the shell region. The angle $\alpha_2$ increases from $-\pi/4$ to $\pi/4$ with the field $H_{\text{tor},2}$. The effective anisotropy in the shell becomes circular ($\alpha_2 = 0$), when $H_{\text{tor},2} = -H_{a,2}\sin 2\psi_2$. The dependence of the effective anisotropy field $H_{\text{eff},2}$ on the field $H_{\text{tor},2}$ has a minimum (see Figure 2b). The minimum attains at $\alpha_2 = 0$, and the minimal value of the effective anisotropy field is equal to $H_{a,2}\cos 2\psi_2$.

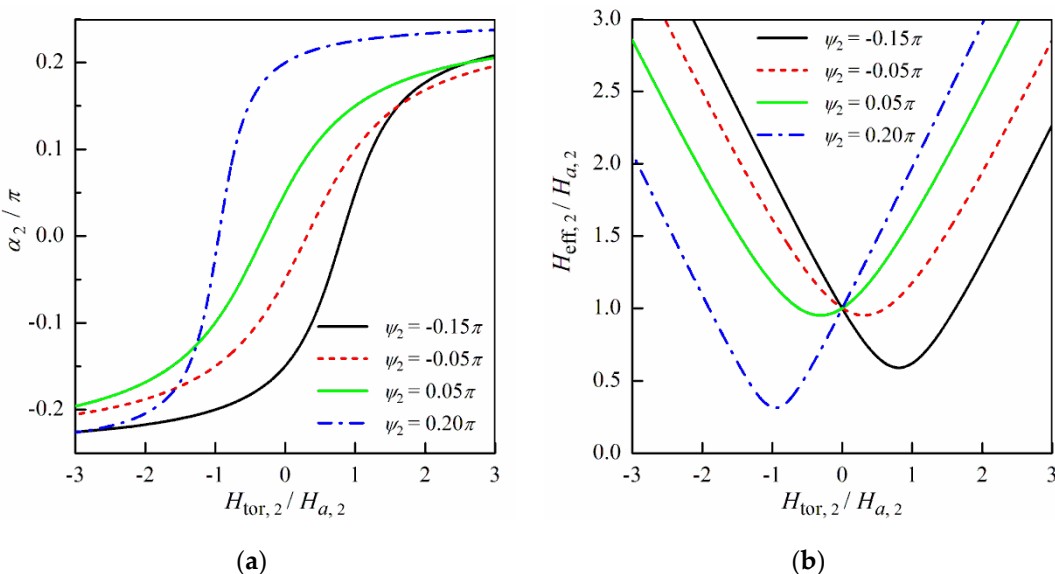

(a)  (b)

**Figure 2.** (a) The effective anisotropy angle $\alpha_2$ and (b) the effective anisotropy field $H_{\text{eff},2}$ in the shell region as a function of the field $H_{\text{tor},2}$ at different values of the intrinsic anisotropy axis angle $\psi_2$.

### 2.2. Impedance Tensor

The GMI effect can be described in terms of the surface impedance tensor [35]. The longitudinal $Z_{zz}$ and off-diagonal $Z_{\varphi z}$ impedance components can be presented as follows [36]:

$$Z_{zz} = (2l/cR)(\zeta_m \sin^2 \theta_2 + \zeta_0 \cos^2 \theta_2),\tag{14}$$

$$Z_{\phi z} = (4\pi N/c)(\zeta_m - \zeta_0)\sin\theta_2\cos\theta_2. \tag{15}$$

Here $l$ is the microwire length, $N$ is the number of turns in the pick-up coil, $\zeta_m$ and $\zeta_0$ are the magnetic and non-magnetic components of the surface impedance tensor and the equilibrium magnetization angle $\theta_2$ in the shell is given by Equation (11).

The surface impedance tensor components $\zeta_m$ and $\zeta_0$ depend on the magnetic structure of the microwire. The expressions for $\zeta_m$ and $\zeta_0$ for the microwire with the core–shell magnetic structure were obtained in [19] assuming that the magnetization in the core region is directed along the microwire axis ($\theta_1 = \pm\pi/2$). However, in the presence of the bias current or the torsional stress, the equilibrium magnetization angle $\theta_1$ deviates from the longitudinal direction (see Equation (5)). Taking into account the deviation of $\theta_1$, the surface impedance component $\zeta_m$ can be expressed as

$$\zeta_m = \frac{ck_2}{4\pi\sigma} \times \frac{J_0(k_2 R) + P_m Y_0(k_2 R)}{J_1(k_2 R) + P_m Y_1(k_2 R)}. \tag{16}$$

Here $J_n$ and $Y_n$ ($n = 0, 1$) are the Bessel functions of the first and the second kind, respectively; $k_2{}^2 = k_0{}^2\mu_2$, $k_0{}^2 = 2i/\delta^2$, $\delta = c/(2\pi\sigma\omega)^{1/2}$, $\sigma$ is the microwire conductivity, $\mu_2$ is the transverse permeability in the shell region and the parameter $P_m$ is given by

$$P_m = \frac{(k_2/k_1)J_0(k_2 r) - QJ_1(k_2 r)}{QY_1(k_2 r) - (k_2/k_1)Y_0(k_2 r)}, \tag{17}$$

where $k_1{}^2 = k_0{}^2\mu_1$, $\mu_1$ is the transverse permeability in the core region and

$$Q = \frac{J_0(k_1 r)}{J_1(k_1 r)}\sin^2\theta_1 + \frac{J_0(k_0 r)}{J_1(k_0 r)}\cos^2\theta_1. \tag{18}$$

Note that at $\theta_1 = \pm\pi/2$ from Equation (18) we have $Q = J_0(k_1 r)/J_1(k_1 r)$, and Equations (16)–(18) transform to expressions obtained in [19]. The non-magnetic component $\zeta_0$ can be obtained from Equations (16) and (17) assuming that $\mu_2 = 1$ [36]. Thus, for $\zeta_0$ we have

$$\zeta_0 = \frac{ck_0}{4\pi\sigma} \times \frac{J_0(k_0 R) + P_0 Y_0(k_0 R)}{J_1(k_0 R) + P_0 Y_1(k_0 R)}, \tag{19}$$

$$P_0 = \frac{(k_0/k_1)J_0(k_0 r) - QJ_1(k_0 r)}{QY_1(k_0 r) - (k_0/k_1)Y_0(k_0 r)}. \tag{20}$$

As follows from Equations (16)–(20) the longitudinal $Z_{zz}$ and off-diagonal $Z_{\varphi z}$ impedance components are governed by the values of the transverse permeability $\mu_j$ in the core and shell regions. The transverse permeability can be found from a solution of the linearized Landau–Lifshitz–Gilbert equation. Taking into account the effects of the bias current and external stresses, we can present the values of $\mu_j$ in the following form [1,21]:

$$\mu_j = \frac{\omega_m^2}{(\omega_m + \omega_j^*)\omega_j^{**} - \omega^2 - i\kappa\omega_m\omega}. \tag{21}$$

Here $\omega_m = \gamma 4\pi M$, $\gamma$ is the gyromagnetic constant, $\kappa$ is the Gilbert damping parameter and

$$\begin{aligned}
\omega_j^* &= \gamma[H_{\mathrm{eff},j}\cos^2(\theta_j - \alpha_j) + H_e\sin\theta_j + H_{b,j}\cos\theta_j], \\
\omega_j^{**} &= \gamma[H_{\mathrm{eff},j}\cos 2(\theta_j - \alpha_j) + H_e\sin\theta_j + H_{b,j}\cos\theta_j].
\end{aligned} \tag{22}$$

The procedure of the calculation of the GMI response in the microwire with the core–shell magnetic structure can be summarized as follows. The equilibrium magnetization angles $\theta_j$ in the core and shell regions can be found from a solution of Equation (11) taking into account expressions (12) and (13). The values of the transverse permeability $\mu_j$ in the core and shell regions are given by Equations (21) and (22). After that, the corresponding

values of $k_1$ and $k_2$ can be calculated, and the impedance components $Z_{zz}$ and $Z_{\varphi z}$ are obtained by means of Equations (14)–(20).

In conclusion of this section, it should be noted that both the magnetization rotation and domain-walls motion contribute to the permeability and magnetoimpedance response. In this work, we analyze the ODMI at sufficiently high frequencies, when the domain-walls motion is strongly damped [1], and the values of the transverse permeability in the core and shell regions are governed by the magnetization rotation only. It has been found that in soft magnetic amorphous materials exhibiting the GMI effect, the domain-walls motion is negligible in the frequency range from several hundred kHz to a few MHz [37–39]. At low frequencies, the contribution of motion of the domain wall between the core and shell to the ODMI response should be taken into account.

## 3. Results

*3.1. Asymmetric Off-Diagonal Magnetoimpedance*

At first, we consider the influence of the bias current on the ODMI in annealed Fe-rich amorphous microwire without external stresses. Note that the field dependence of the ODMI response $Z_{\varphi z} = R_{\varphi z} - iX_{\varphi z}$ (here $R_{\varphi z}$ and $X_{\varphi z}$ are the real and imaginary parts of the ODMI) was described previously for a microwire with the core–shell structure assuming that the magnetization in the core region has the longitudinal direction [19]. To obtain the ODMI effect with high field sensitivity, the domain structure in the shell region should be removed. The threshold field $H_{th}$ of the bias current to eliminate the surface domain structure can be expressed as follows [21,40]: $H_{th} = H_{a,2} |\sin \psi_2|$. Taking into account Equation (7), we find for the threshold bias current $I_{th}$:

$$I_{th} = (cRH_{a,2}/2)|\sin \psi_2|. \tag{23}$$

Assuming for estimations that $2R = 15$ μm, $\psi_2 = -0.05\pi$ and $H_{a,2} = 30$ Oe, we find for the value of the threshold bias current $I_{th} \cong 17.5$ mA.

Shown in Figure 3 are the field dependences of the real $R_{\varphi z}$ and imaginary parts $X_{\varphi z}$ of the ODMI calculated for different values of $I_b > I_{th}$. For convenience, the values of $R_{\varphi z}$ and $X_{\varphi z}$ are reduced to the characteristic off-diagonal impedance $Z_0$:

$$Z_0 = 2\pi N R Z_{DC}/l = 2N/\sigma R, \tag{24}$$

where $Z_{DC} = l/\pi\sigma R^2$ is the resistance in the direct current mode. Assuming that $N = 50$, $2R = 15$ μm and $\sigma = 5 \times 10^{15}$ s$^{-1}$, we obtain $Z_0 \cong 24$ Ohm.

At $I_b > I_{th}$, the asymmetry in the field dependence of the ODMI appears The asymmetry arises due to the interaction of the helical anisotropy with the circular magnetic field induced by the bias current [1,21,40]. The real and imaginary parts of the ODMI increase sharply at $I_b \cong I_{th}$. With a further increase of the bias current, the field sensitivities of $R_{\varphi z}$ and $X_{\varphi z}$ decrease due to a drop in the transverse permeability in the shell region, however, they remain sufficiently high within a wide range of the bias current.

Although the shell region with a helical anisotropy makes the main contribution to the ODMI of the microwire, the ODMI is very sensitive to the volume part of the core region. Figure 4 illustrates the effect of the core region diameter $2r$ on the ODMI. The dependences of $R_{\varphi z}$ and $X_{\varphi z}$ on the external field have similar behavior for all values of $2r$, and the decrease of the core diameter results in a growth of the ODMI response. Note that the volume parts of the core and shell can be tuned by stress-annealing. In particular, an increase in the annealing temperature and time [12] and tensile stress during the annealing [14,41,42] leads to a decrease in the core region volume.

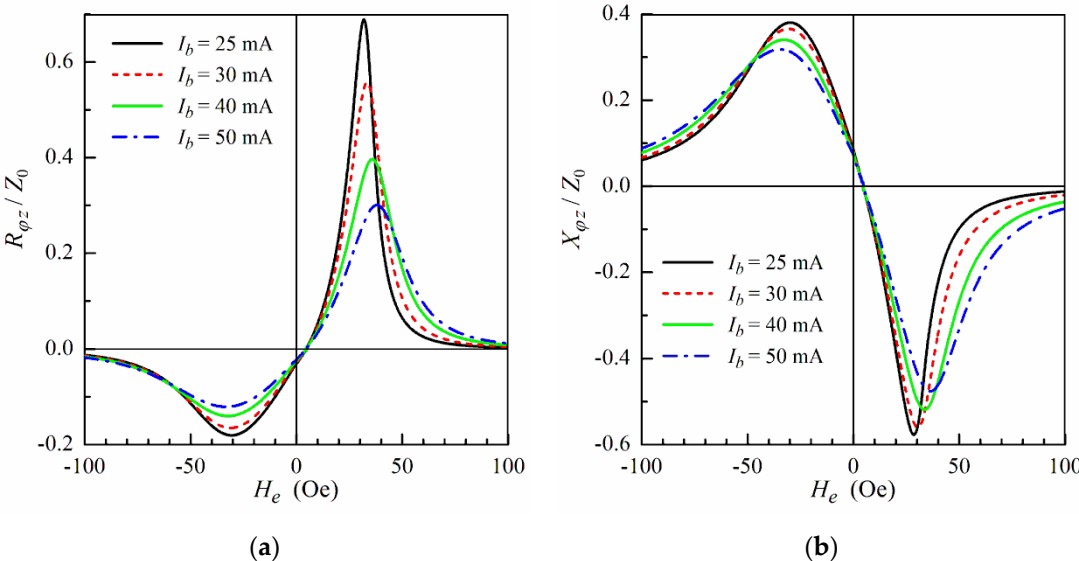

**Figure 3.** (**a**) The real part $R_{\varphi z}$ and (**b**) the imaginary part $X_{\varphi z}$ of the ODMI as a function of the external field $H_e$ at $f = \omega/2\pi = 100$ MHz for different values of the bias current $I_b$. Parameters used for calculations are $2R = 15$ μm, $2r = 8$ μm, $M = 900$ G, $\sigma = 5 \times 10^{15}$ s$^{-1}$, $\kappa = 0.15$, $H_{a,1} = 5$ Oe, $H_{a,2} = 30$ Oe and $\psi_2 = -0.05\pi$.

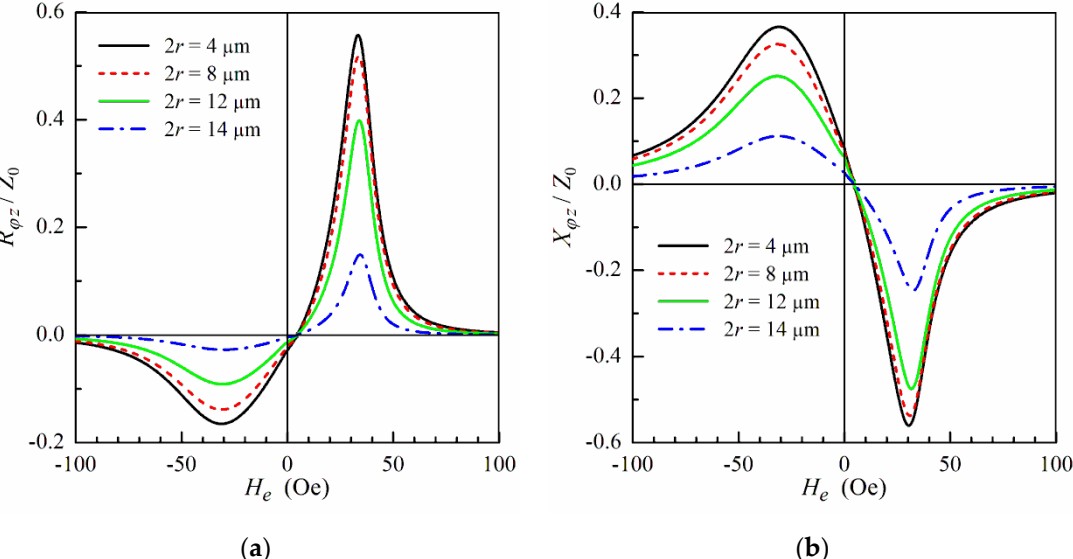

**Figure 4.** (**a**) The real part $R_{\varphi z}$ and (**b**) the imaginary part $X_{\varphi z}$ of the ODMI as a function of the external field $H_e$ at $f = 100$ MHz and $I_b = 30$ mA for different values of the core region diameter $2r$. Other parameters used for calculations are the same as in Figure 3.

It follows from Figures 3 and 4 that the field dependences of the real and imaginary parts of the ODMI response show nearly linear behavior at $H_e \cong -H_{a,2}\sin 2\psi_2$, where the ODMI turns to zero. To analyze the frequency dependences of $R_{\varphi z}$ and $X_{\varphi z}$, let us introduce the field sensitivities of the real $S_R$ and imaginary $S_X$ parts of the ODMI defined as follows [43]:

$$S_R = R_p/\Delta H, \tag{25}$$

$$S_X = |X_p|/\Delta H. \tag{26}$$

Here $R_p$ and $|X_p|$ are the maximum values of $R_{\varphi z}$ and $|X_{\varphi z}|$ at positive external fields, $\Delta H = H_p + H_{a,2}\sin 2\psi_2$ and $H_p$ is the external field at the peak.

The frequency dependences of the field sensitivities $S_R$ and $S_X$ calculated for different values of the bias current are presented in Figure 5. The values of $S_R$ and $S_X$ have different frequency behavior. The field sensitivity $S_R$ of the real part of the ODMI increases monotonically with the frequency, whereas $S_X$ has a maximum at a certain frequency. At low frequencies, the field sensitivity $S_X$ is higher than $S_R$. It also follows from Figure 5 that field sensitivities decrease with a growth of the bias current $I_b$. However, the magnitudes of $S_R$ and $S_X$ are relatively high within a wide range of the bias current.

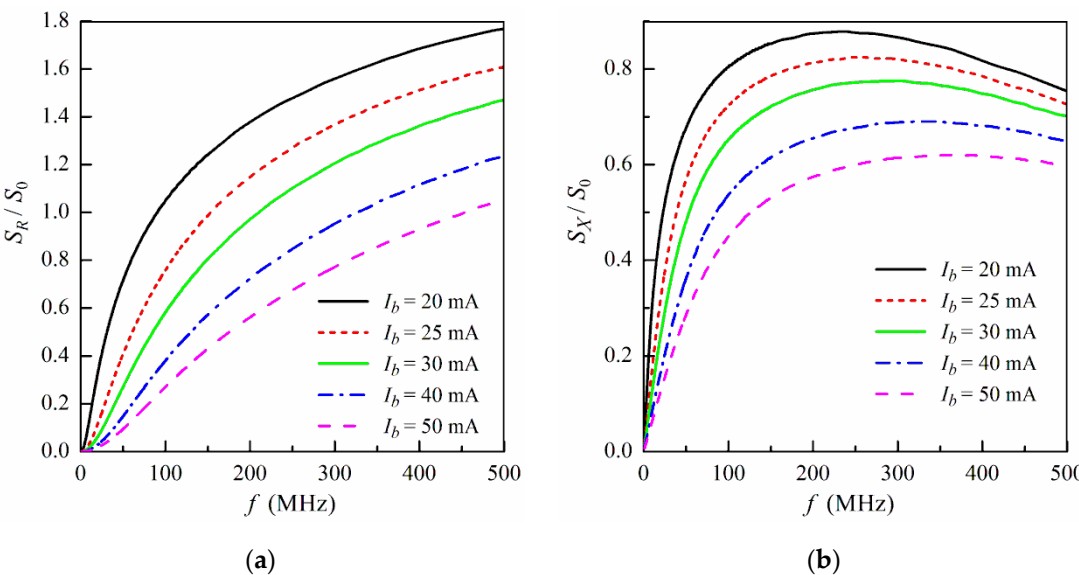

(**a**)                                                                                  (**b**)

**Figure 5.** The field sensitivity of the real part $S_R$ (**a**) and imaginary part $S_X$ (**b**) of the ODMI as a function of the frequency $f$ for different values of the bias current $I_b$. Other parameters used for calculations are the same as in Figure 3. The values of $S_R$ and $S_X$ are reduced to the characteristic field sensitivity $S_0 = Z_0/H_{a,2}$.

Note that the obtained field dependences of the ODMI in the presence of the bias current describe qualitative results of the experimental studies of the OMDI effect in stress-annealed Fe-rich microwires [11–13]. The origin of the strong ODMI is related to a helical anisotropy in the shell region of the microwire, which appears after the annealing. Although the main contribution to the ODMI comes from the shell, the core region affects the ODMI through its volume and a deviation of the magnetization from the longitudinal direction in the inner part of the microwire.

### 3.2. Effect of Tensile Stress on Off-Diagonal Magnetoimpedance

The application of the tensile stress to the microwire results in changes in the equilibrium magnetization distribution and consequently affects the ODMI response. Figure 6 shows the field dependences of $R_{\varphi z}$ and $X_{\varphi z}$ calculated for different values of the tensile stress $\sigma_t$. Both the real and imaginary parts of the ODMI exhibit similar behavior with an increase in stress. The peaks in $R_{\varphi z}$ and $X_{\varphi z}$ become more pronounced, and the peak fields shift towards zero fields. It follows from Figure 6 also that the field sensitivity of the ODMI increases with the tensile stress.

The evolution of the ODMI response in the presence of tensile stress is related to the changes in the effective anisotropy in the microwire shell region. With an increase of the tensile stress, the effective anisotropy angle $\alpha_2$ in the shell deviates from the azimuthal direction, and the effective anisotropy field $H_{eff,2}$ decreases (see Equations (12) and (13)). Taking into account the changes in $\alpha_2$ and $H_{eff,2}$ under the effect of the tensile stress, we can present the expression for threshold field $H_{th}$ of the bias current to eliminate domain

structure in the shell region in the following form: $H_{th} = H_{eff,2}|\sin\alpha_2|$. Correspondingly, the expression for the threshold bias current $I_{th}$ can be rewritten as

$$I_{th} = (cRH_{eff,2}/2)|\sin\alpha_2|. \tag{27}$$

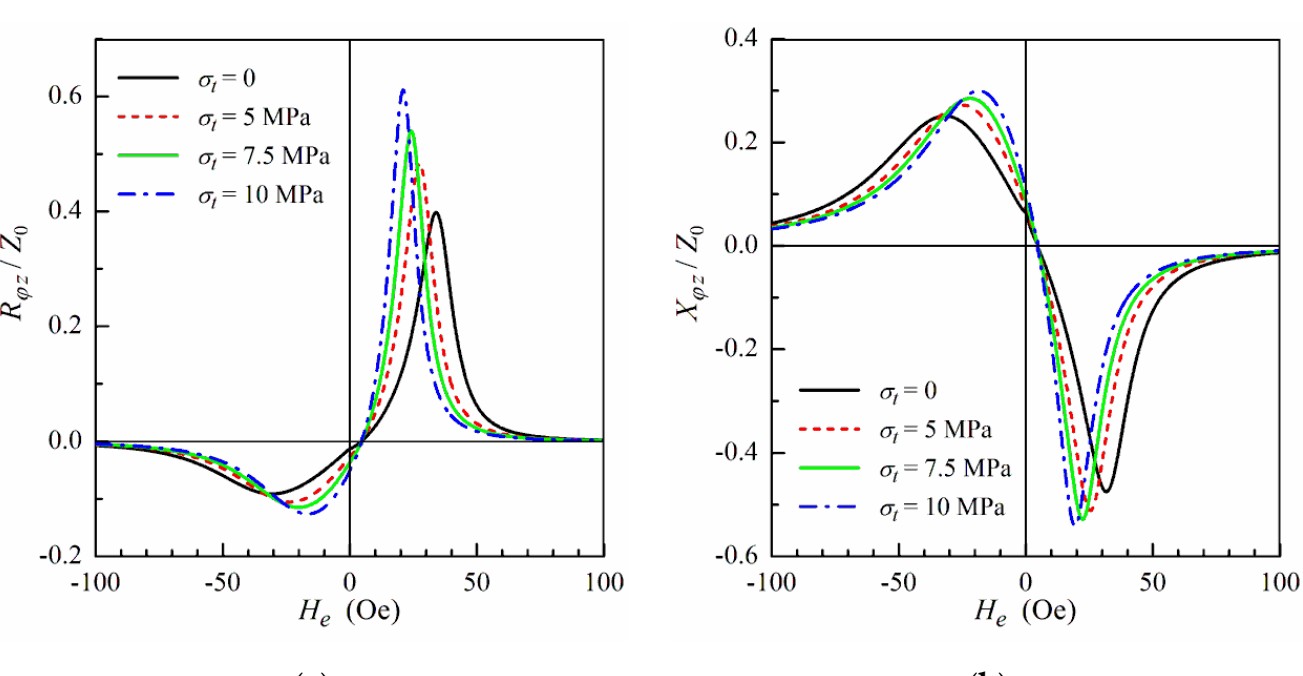

**Figure 6.** (**a**) The real part $R_{\varphi z}$ and (**b**) the imaginary part $X_{\varphi z}$ of the ODMI as a function of the external field $H_e$ at $f$ = 100 MHz and $I_b$ = 30 mA for different values of the tensile stress $\sigma_t$. Magnetostriction coefficient $\lambda_s = 40 \times 10^{-6}$, other parameters used for calculations are the same as in Figure 3.

It follows from Equations (12), (13), and (27) that the threshold current increases with the tensile stress. Hence, the threshold current $I_{th}$ tends to the bias current $I_b$ under the application of the tensile stress. As discussed above, the ODMI increases when the bias current is close to $I_{th}$ (see Figure 3). Thus, an increase of the tensile stress $\sigma_t$ results in a growth of the ODMI response (see Figure 6).

The frequency dependences of the field sensitivities $S_R$ and $S_X$ for different values of the tensile stress $\sigma_t$ are shown in Figure 7. The values of $S_R$ and $S_X$ are calculated by means of Equations (25) and (26) taking into account that $\Delta H = H_p + H_{eff,2}\sin 2\alpha_2$. The field sensitivities increase monotonically with the value of the applied stress (see Figure 7).

To describe the ODMI sensitivity to the tensile stress $\sigma_t$ we introduce the ratios

$$(\Delta R/R_{\varphi z})_\sigma = [R_{\varphi z}(\sigma_t) - R_{\varphi z}(0)]/R_{\varphi z}(0), \tag{28}$$

$$(\Delta X/R_{\varphi z})_\sigma = [X_{\varphi z}(\sigma_t) - X_{\varphi z}(0)]/X_{\varphi z}(0), \tag{29}$$

where $R_{\varphi z}(0)$ and $X_{\varphi z}(0)$ are the real and imaginary parts of the ODMI without tensile stress.

The frequency dependences of $(\Delta R/R_{\varphi z})_\sigma$ and $(\Delta X/X_{\varphi z})_\sigma$ calculated for a fixed external field $H_e < H_p$ at $f$ = 100 MHz for different values of the bias current are shown in Figure 8. Both the ratios of $(\Delta R/R_{\varphi z})_\sigma$ and $(\Delta X/X_{\varphi z})_\sigma$ have high sensitivity to the applied tension stress. As follows from Figure 8, the ratios of $(\Delta R/R_{\varphi z})_\sigma$ and $(\Delta X/X_{\varphi z})_\sigma$ exhibit nearly linear dependence on the tensile stress at low values of $\sigma_t$. This fact is attractive for the development of stress sensors.

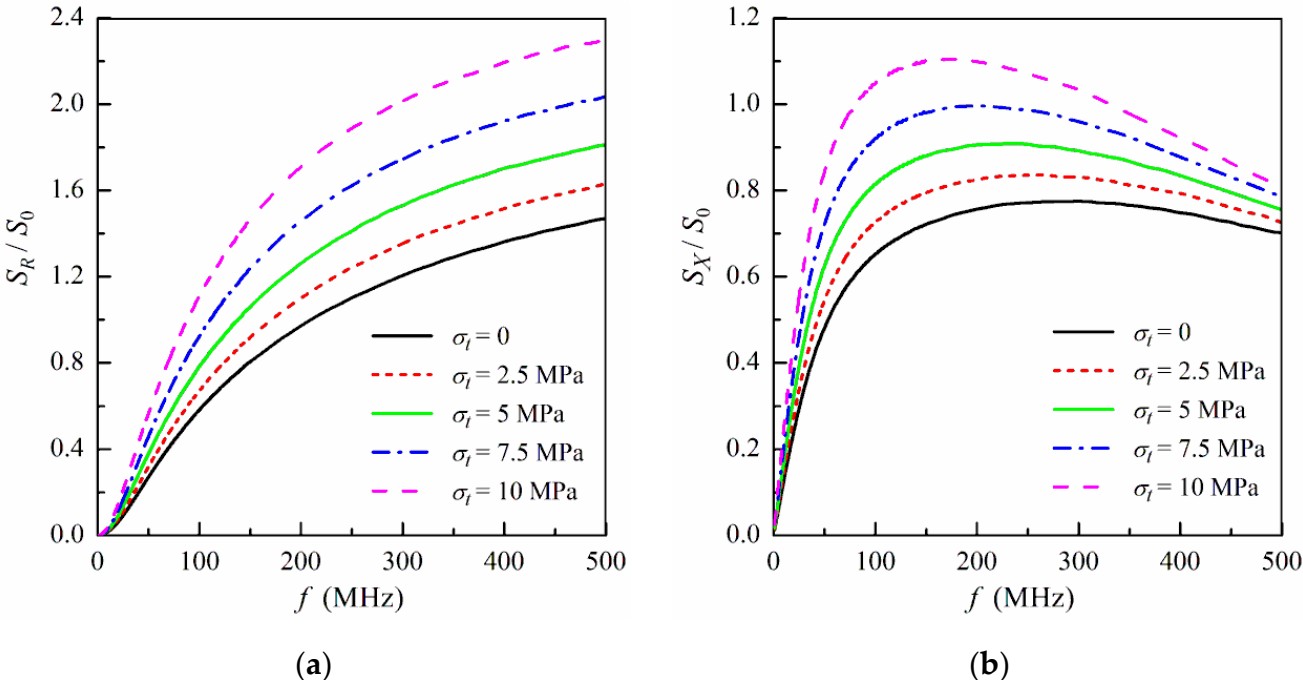

**Figure 7.** The field sensitivity of the real part $S_R$ (**a**) and imaginary part $S_X$ (**b**) of the ODMI as a function of the frequency $f$ at $I_b$ = 30 mA for different values of the tensile stress $\sigma_t$. Magnetostriction coefficient $\lambda_s = 40 \times 10^{-6}$, other parameters used for calculations are the same as in Figure 3.

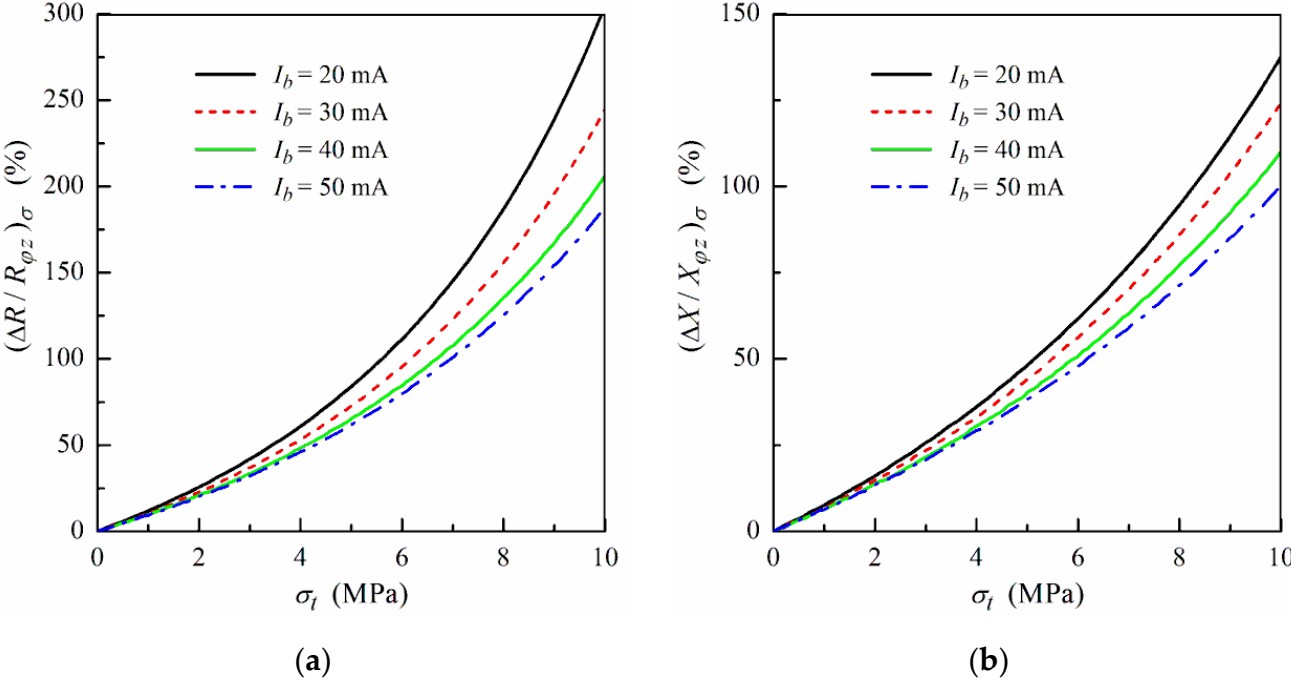

**Figure 8.** The ratio of $(\Delta R/R_{\varphi z})_\sigma$ (**a**) and $(\Delta X/X_{\varphi z})_\sigma$ (**b**) as a function of the tensile stress $\sigma_t$ at $H_e$ = 10 Oe and $f$ = 100 MHz for different values of the bias current $I_b$. Magnetostriction coefficient $\lambda_s = 40 \times 10^{-6}$, other parameters used for calculations are the same as in Figure 3.

### 3.3. Effect of Torsional Stress on Off-Diagonal Magnetoimpedance

The influence of the torsional stress on the ODMI response is more complicated. The application of the torsional stress to the microwire induces a helical anisotropy in the whole sample. Depending on the direction of the torsional stress, the anisotropy axis angle in the shell region can enhance or tend to zero (see Figure 2a). In addition, the

torsional stress deviates slightly from the easy magnetization axis in the core region from the longitudinal direction.

The dependences of the real $R_{\varphi z}$ and imaginary parts $X_{\varphi z}$ of the ODMI on the external field are shown in Figure 9 at fixed $I_b$ and different values of the angular displacement per unit length $\tau > 0$. For low $\tau$, the asymmetry between the absolute values of the peaks in $R_{\varphi z}$ and $X_{\varphi z}$ at positive and negative fields decreases. At some critical value of $\tau = \tau_{cr}$, the absolute values of the peaks become equal. At $\tau > \tau_{cr}$, the modulus of the peak in the real and imaginary parts of the ODMI at a negative external field becomes higher (see Figure 9). The critical value $\tau_{cr}$ of the angular displacement per unit length corresponds to the appearance of the effective circular anisotropy in the shell region ($\alpha_2 = 0$). It follows from Equation (12) that this condition satisfies when $H_{tor,2} = -H_{a,2}\sin 2\psi_2$. Taking into account Equation (10), we obtain for $\tau_{cr}$ [44]:

$$\tau_{cr} = -MH_{a,2}\sin 2\psi_2 / 3\lambda_s GR. \tag{30}$$

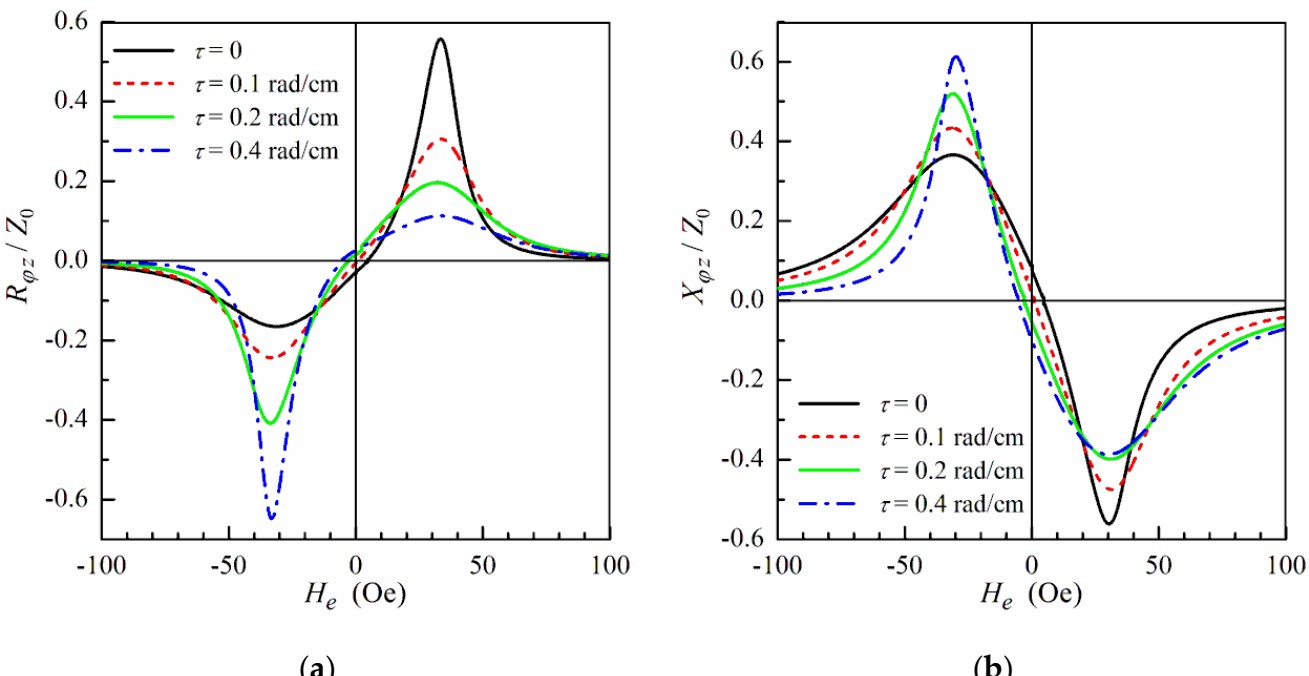

**Figure 9.** (**a**) The real part $R_{\varphi z}$ and (**b**) the imaginary part $X_{\varphi z}$ of the ODMI as a function of the external field $H_e$ at $f$ = 100 MHz and $I_b$ = 30 mA for different values of the angular displacement per unit length $\tau$. Magnetostriction coefficient $\lambda_s = 40 \times 10^{-6}$, shear modulus $G$ = 50 GPa, other parameters used for calculations are the same as in Figure 3.

For negative torsional stresses ($\tau < 0$), the increase of the absolute value of $\tau$ leads to a growth of the effective anisotropy field $H_{eff,2}$ and effective anisotropy angle modulus in the shell. As a result, the threshold bias current $I_{th}$ increases (see Equation (27)). Correspondingly, the peaks in $R_{\varphi z}$ and $X_{\varphi z}$ at negative external field growth. The maximal field sensitivity of the ODMI is achieved when the bias current $I_b$ tends to the threshold one $I_{th}$. With a further increase of the stress absolute value, the bias current becomes less than $I_{th}$, and the field dependences of $R_{\varphi z}$ and $X_{\varphi z}$ exhibit hysteretic behavior [44].

Figure 10 shows the frequency dependences of the field sensitivities of the real and imaginary parts of the ODMI calculated for different values of the angular displacement per unit length $\tau$. At not-too-low frequencies, the field sensitivity $S_R$ of the real part of the ODMI is higher than $S_X$. Both the sensitivities $S_R$ and $S_X$ decrease monotonically with an increase of the angular displacement per unit length $\tau$.

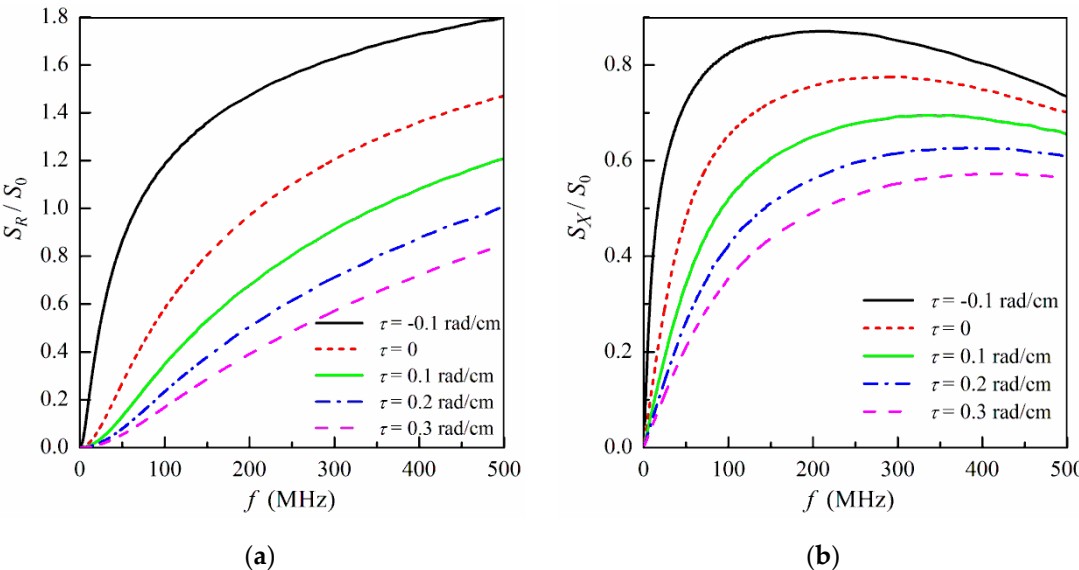

**Figure 10.** The field sensitivity of the real part $S_R$ (**a**) and imaginary part $S_X$ (**b**) of the ODMI as a function of the frequency $f$ at $I_b$ = 30 mA for different values of the angular displacement per unit length $\tau$. Magnetostriction coefficient $\lambda_s = 40 \times 10^{-6}$, shear modulus $G$ = 50 GPa, other parameters used for calculations are the same as in Figure 3.

## 4. Discussion

Fe-rich glass-coated amorphous microwires with positive magnetostriction are attractive for GMI applications due to higher saturation magnetization and lower price in comparison with Co-rich microwires. However, as-prepared Fe-rich microwires exhibit a low GMI effect. It was found that stress-annealing or Joule heating may enhance significantly the GMI in amorphous microwires with positive magnetostriction [8,17]. The improvement of the GMI effect is related to changes in the distribution of the residual stresses within Fe-rich microwire after annealing resulting in the appearance of circular or helical magnetic anisotropy at the microwire surface layer. Recently, an electrodynamic model was proposed to describe the GMI effect in stress-annealed amorphous microwires with positive magnetostriction [19]. The obtained theoretical dependences allowed one to explain the main results of experimental studies of the GMI in stress-annealed Fe-rich amorphous microwires. Using the anisotropy fields in the core $H_{a,1}$ and shell $H_{a,2}$, the anisotropy axis angle in the shell $\psi_2$, and the core diameter $2r$ as fitting parameters, the calculated results explained the evolution of the field dependence of the GMI response with a frequency increase (the transition from the single-peak to two-peak behavior). In addition, the theoretical results described the positions of the peaks in the GMI field dependence and the magnitude of the GMI ratio.

In this work, we modify the model to analyze the effect of tensile and torsional stresses on the ODMI in annealed Fe-rich glass-coated amorphous microwires. The stress application changes the magnetization distribution in the microwire due to the interaction of the intrinsic magnetic anisotropy and the magnetoelastic anisotropy induced by external stress. The results of modeling show that the ODMI response in annealed Fe-rich microwire can be enhanced by tensile or torsional stress. It is demonstrated that the main contribution to the ODMI in the microwire comes from the surface region with a helical anisotropy. However, the central region of the microwire can influence the ODMI response through its volume and a deviation of the magnetization from the longitudinal direction.

It is well-known that the strong ODMI effect in amorphous microwires can be observed when the bias current is applied to the sample to remove a domain structure in the microwire surface region. However, high values of the bias current results in Joule heating. It can lead to crystallization and degradation of the soft magnetic properties of

the microwire [7,12]. Correspondingly, there are some restrictions on the use of the ODMI effect in technological applications.

In this work, we analyze the ODMI effect in annealed Fe-rich amorphous microwires with positive magnetostiction. Note that the approach proposed allows one to describe also the influence of the tension and torsional stresses on the longitudinal GMI in microwires with the core–shell magnetic structure.

In conclusion of this section, it should be noted that there are no experimental studies of the ODMI in annealed amorphous microwires with positive magnetostriction in the presence of external stresses, and further verification of predictions of the proposed model is required. However, the effect of the external stresses on the ODMI seems to be promising for applications. The obtained results demonstrate that the field dependence of the ODMI is very sensitive to external stresses, and it may allow one to improve the sensitivity of the magnetic-field sensors.

## 5. Conclusions

The influence of the tensile and torsional stresses on the ODMI in annealed glass-coated amorphous microwires with positive magnetostrcition is analyzed by means of the core–shell magnetic structure. It is assumed that the microwire has an axial anisotropy in the inner core and a helical anisotropy in the external shell region, which occurs after the annealing. The effect of the applied stresses on the ODMI is related to the interaction of the microwire internal anisotropy and the magnetoelastic anisotropy induced by external stresses. The microwire impedance tensor is obtained taking into account the magnetoelastic anisotropy. It is demonstrated that the external stresses affect the ODMI response and may enhance the ODMI field sensitivity within a wide frequency range. The obtained results may be useful from the point of view of the development of the magnetic field and stress sensors.

**Funding:** This research received no external funding.

**Institutional Review Board Statement:** Not applicable.

**Informed Consent Statement:** Not applicable.

**Data Availability Statement:** The data that support the findings of this study are available from the author upon reasonable request.

**Acknowledgments:** The author would like to thank Serghei Baranov and Vyacheslav Popov for fruitful discussions.

**Conflicts of Interest:** The author declares no conflict of interest.

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
