# Peer review of "Off-Diagonal Magnetoimpedance in Annealed Amorphous Microwires with Positive Magnetostriction: Effect of External Stresses"

_2673-8724, doi:10.3390/magnetism3010005_

Round 1

Reviewer 1 Report

The author studied theoretically the influence of the external stresses on the off-diagonal magnetoimpedance in annealed amorphous microwires with positive magnetostriction and the simulation results with different parameters were given. But there exist serious problems that should be addressed and also there are modifications that should be corrected as follow to improve the paper.

1. In the framework of the model, the authors neglect the exchange and magnetostatic coupling between the two regions of the microwire to simplify calculations. This may be a reason behind the difference between experimental and calculated results.

2. The authors give a large number of model predictions. However, there is a lack of experimental verification of the predicted results, which is very necessary.

And some general points:

3.     There are incorrect characters in Figure 1(a).

4.     It is recommended that the figure legends are not represented by numbers, it needs to specify parameter values.

5.     Please show the theoretical results to compare with the known experimental results.

Reviewer 2 Report

This paper gives theoretical report on the influence of the tensile and torsional stresses on the off-diagonal magnetoimpedance (ODMI) in annealed glass-coated amorphous microwires in the framework of the core-shell magnetic structure. It is indicated that the external stresses affect the ODMI, giving a possibility for the development of magnetic-field and stress sensors.

The assumed model and calculation method seem reasonable and reliable, and the influence of the stresses are phenomenologically well described.  I basically recommend publication of this paper, but it would be desirable to compare the results obtained at least qualitatively with experimental reports. In addition I have a question: is there any adjustable parameters to in order to reproduce the experimental results?

Reviewer 3 Report

The author has provided an extension of their prior work to include the effects of isotropic Joulian magnetostriction on the impedance of a core-shell wire. Overall the paper is well written and easy to follow. However, some questions about the accuracy of the magnetostriction calculations, and significance of the dynamic results needs to be asked.

Most of my questions stem from the initial description of the model being a bit vague, even after looking back through your 2018 and 2021 papers that this builds on. A simple schematic of the wire, with your coordinates properly labeled would be immensely helpful. The next two paragraphs describe my current understanding of your setup. They could both simply be me misunderstanding your setup, but if I'm reading things correctly, then there are some rather large flaws with how some of the energies are being described.

Noticeably, there are only two angles included in the equations, with theta being the magnetization angle, and psi the anisotropy angle, both with respect to the azimuthal direction. For this setup I would assume you would need three direction 1) the anisotropy direction, and 2/3) the orientation of the magnetization in spherical coordinates. Your description seems to mean theta = 0 is a circular magnetization field (i.e., always parallel to the azimuthal basis vector in polar coordinates), while theta = pi/2 would be a magnetization field parallel to the radial direction. If we apply that same logic to psi, the description seems to be flawed. You include the statement that for the core, which has a uniaxial anisotropy, that psi = pi/2. Based on the description you've provided that would be a radial anisotropy, not uniaxial. A uniaxial anisotropy, with the longitudinal axis being the unique axis, would produce an isotropic easy or hard plane for the wire's cross section. For an isotropic plane the orientation of the magnetization within that plane wouldn't change the energy at all. The energy wouldn't depend on that angle.

If the above statements are true, then it's problematic that the same angle theta is used for the tensile magnetostrictive energy term. For isotropic magnetostriction, the application of uniaxial tension (compression) to the wire will result in the longitudinal axis becoming the easy (hard) axis (i.e., indistinguishable from the uniaxial anisotropy described above). So just as above, the plane containing the wire's cross sectional area would become an isotropic hard (easy) plane / disc. In other words, the effect of uniaxial tension (compression) has zero dependence on the azimuthal orientation of the magnetization. That energy would depend exclusively on the the 'elevation' angle of the magnetization (i.e., the orientation w.r.t. the longitudinal direction, not in the plane perpendicular to it). Please note how much easier the last two paragraphs would be to write if you had included a schematic outlining your geometry in the paper. 

In addition to the points above, there are some smaller points of concern I would still have, even if my statements above are simply a misunderstanding of the author's work:

Provide a citation for equation 2. Shouldn't that term also depend on the angle theta, seeing as it's a Zeeman energy? What assumptions have you made to simplify that?

The assumption on lines 144-146 (page 4): Why are these assumptions being made? It would be thermodynamically consistent with the prior assumptions to integrate the energy density through the radial direction, and then minimize the resulting energy which would be a function of the magnetic orientation. In other words, the magnetization distribution will minimize the total energy of the system, not the energy density locally. Since both energy densities are linear in rho, this should be easily accomplished. 

The dynamic results would likely be strongly affected when the AC current frequency overlaps with the resonant frequency of the domain wall separating the core / shell regions. However, the assumption of zero exchange coupling between core / shell, and in general maintaining the boundary at rho =r as a constant, instead of a dynamic property, prohibits your model from capturing this effect. Please provide a comment addressing this limitation. 

Most of your figures have plots with 4 to 5 lines, where the meaning of the numbers 1-4 (or 5) changes from figure to figure. Please update the legends to be more descriptive. For example, figures 1 and 2 of your 2021 paper clearly indicate what each line means in the legend. That is much nicer from the reader's perspective, instead of needing to hunt for the details in the figure caption.

In figure 1a, the y-axis is labeled as  psi_2 however the caption indicates it should be alpha_2. 

Line 169 "at high negative direction of the stress" - do you mean 'compression'? Similar question for the next line, does "at high positive stress" mean tension? Please note that placing a wire in a meaningful amount of compression is all but impossible without it buckling. However, your statements about compression would be relevant for a negative magnetostrictive material in tension. That would be a slightly more worthwhile comment. 

For most of your impedance equations you reference the reader to citation 19. However, in that paper you simply reference those equations from even earlier work. Please provide a relevant citation that shows where those equations were derived, not simply the last place you used them. 

Reviewer 4 Report

The manuscript “Off-Diagonal Magnetoimpedance in Annealed Amorphous

Microwires with Positive Magnetostriction: Effect of External Stresses” by Nikita A. Buznikov is a theoretical investigation of the stress effect on giant magnetoimpedance on microwires with positive magnetostriction following a previous publication by the same author which proposes a core shell model for this type of microwires.

The article is well written and the model is clearly explained. The predictions of the model are especially important for the development of stress sensors but also for general applications in which external stress may act on the microwires with positive magnetostriction.

Though the experimental validation is not yet possible,  as to my knowledge there are no experimental results to compare with, the results presented in the manuscript are novel and important as it could facilitate and motivate further experimental investigations on this important class of materials. Therefore, I recommend that the manuscript is published as it is.

Round 2

Reviewer 1 Report

Although the paper has been modified and explained a lot, it is necessary to verify the theoretical model. The theoretical model lacks some validation.

Reviewer 2 Report

Now that I understand that there are no experimental studies of the magnetoimpedance in annealed amorphous microwires, and my previous question can be rewritten as follows: 

If the experimental results would be obtained in future, is it possible to optimize the parameters in order to fit the calculation to the experimental results?

I recommend the publication of this paper after adding the answer to the above simple question. 

Reviewer 3 Report

Thank you for the inclusion of figure 1 depicting the wire geometry and variables used in the model. It greatly clarifies the resulting model. For future reference, having angles defined within in the phi-z plane is a non-standard description when using cylindrical coordinates. That's what lead to all my confusion with the first draft of the paper. 

A comment made in your response provides an unstated assumption. You assume the magnetization is constrained to the phi-z plane. That's why the magnetic moment only has one degree of freedom (theta_j) instead of two. While it's clear from the equations, it would be worth adding a statement clearly specifying this.

Concerning comment 3 on your assumptions for minimizing the free energy:

I agree with you that the key difference between your existing approach, and the proposed integration w.r.t. rho will be numerical factors that change the magnitude of some of the fields introduced in equations 6-10, while the overall equations / predictions will retain the same form. Please provide a statement to that extent in the paper. That way it is clear that future improvements upon your model, potentially by other authors, can / should be cast back into the simplified form you present here.

Concerning the dynamic predictions of the model: 

It's fine to neglect wall motion under the assumption that you are operating at high enough frequencies that the motion is damped out. However, in that case it seems quite odd to have all your frequency plots from only 0-500 MHz (i.e., including a range where wall motion could dominate the frequency response). What are the FMR frequencies for the core/shell being simulated? It appears all of your plots are below FMR seeing as the real part of the sensitivity S_R is monotonically increasing. Is there a reason you're not showing the frequency response through FMR?

Finally, most reviewers have asked how this model compares to experimental data, to which you've responded that no data exists. That's not entirely true. You have cited multiple papers with an abundance of experimental data. You seem to be avoiding comparison to those models as they were tested in a `stress free' configuration. However, 1) there is certainly residual stress in those wires, and 2) your model should still be required to work when the stress is zero! So you can compare to the experimental literature, however in that case the stress values in your model may become fit parameters.

From my perspective it seems to be a requirement for you to provide a comparison to experimental data in order for this work to have any impact. Otherwise the large assumptions you've made in neglecting exchange and magnetostatic coupling are too large to ignore, especially as some of your results appear to be extrapolating outside of the regime where your assumptions are valid. In general, I appreciate your desire to keep the model as simple as possible, but in that case you need to clearly show that the model still works by providing at least some level of validation. Otherwise it may be simplified to the point of irrelevancy. 

Round 3

Reviewer 1 Report

These issues have been totally revised. I believe the manuscript has been sufficiently improved to warrant publication in Magnetism.

Reviewer 3 Report

With the recent additions, the authors have fully addressed all my comments. After a further review I also agree with the authors assertion that they are managing the dynamics / frequency response of this system in an appropriate manner. In my prior review it appears I had made a typo in a literal back of the envelope calculation for the skin depth, which had led to some of my confusion over the relevant frequency scales in your problem. The new paragraph you included nicely explains those issues with relevant models and experiments.